# ELoran Propagation Delay Prediction Model Based on a BP Neural Network for a Complex Meteorological Environment

**DOI:** 10.3390/s23115176

**Published:** 2023-05-29

**Authors:** Shiyao Liu, Wei Guo, Yu Hua, Wudian Kou

**Affiliations:** 1National Time Service Center, Chinese Academy of Sciences, Xi’an 710600, China; 2Key Laboratory of Precise Positioning and Timing Technology, Chinese Academy of Sciences, Xi’an 710600, China

**Keywords:** eLoran, meteorological factor, propagation delay prediction model, Back-Propagation neural network

## Abstract

The core of eLoran ground-based timing navigation systems is the accurate measurement of groundwave propagation delay. However, meteorological changes will disturb the conductive characteristic factors along the groundwave propagation path, especially for a complex terrestrial propagation environment, and may even lead to microsecond-level propagation delay fluctuation, seriously affecting the timing accuracy of the system. Aiming at this problem, this paper proposes a propagation delay prediction model based on a Back-Propagation neural network (BPNN) for a complex meteorological environment, which realizes the function of directly mapping propagation delay fluctuation through meteorological factors. First, the theoretical influence of meteorological factors on each component of propagation delay is analyzed based on calculation parameters. Then, through the correlation analysis of the measured data, the complex relationship between the seven main meteorological factors and the propagation delay, as well as their regional differences, are demonstrated. Finally, a BPNN prediction model considering regional changes of multiple meteorological factors is proposed, and the validity of the model is verified by long-term collected data. Experimental results show that the proposed model can effectively predict the propagation delay fluctuation in the next few days, and its overall performance is significantly improved compared with that of the existing linear model and simple neural network model.

## 1. Introduction

The Loran-C system is a low-frequency (LF) ground-based navigation system developed by the US Navy for military purposes in the 1950s based on the hyperbolic positioning principle [1,2]. The enhanced Loran (eLoran) system is an upgraded version of the Loran-C system. With many advantages such as wide coverage, high stability and strong anti-interference ability, the eLoran system has become a reliable backup for global navigation satellite systems (GNSS) and will be combined with GNSS to further improve the national satellite-terrestrial integrated positioning, navigation and timing (PNT) system [3,4].

In order to improve the timing accuracy of the eLoran system, it is necessary to accurately measure the propagation delay. However, the propagation delay is time-varying, which will be analyzed in detail in the following sections. Currently, the most effective method is to aggregate the fluctuations caused by various factors into the ASF fluctuation term, develop a high-precision ASF grid map and provide delay correction services for users through ASF interpolation [5,6]. The basis of this method is the precise prediction of the signal propagation delay and ASF correction values of grid vertices. Fluctuations of various meteorological factors such as temperature, humidity, precipitation, etc. will cause different changes in conductive characteristics along the propagation path and may lead to accumulated propagation delay measurement errors, even up to the microsecond level, seriously affecting the accuracy of ASF correction [7]. Therefore, studying the impact of various meteorological factors on eLoran signal propagation delay fluctuation is the key to accurately establishing a propagation delay prediction model, which is of great significance for improving the service quality of the eLoran system.

In this study, the accuracy of meteorological data acquisition is the basis of propagation delay prediction, and the development of remote sensing technology is the core of meteorological element data acquisition. John Houghton, a climatologist and pioneer in the field, has made outstanding contributions to the study of climate change and the physical sciences of the atmosphere. Sir Houghton cut his teeth in the 1970s designing instruments for measuring the atmospheric temperature and composition for NASA’s four rain-cloud satellites and went on to lead his team in developing a range of remote sensing instruments for ocean climate measurements. His contributions include, but are not limited to, the foundation of remote sensing, especially in early satellite technology, the selective chopper radiometer, the atmospheric sounding, the clouds classification, the weather satellites and the radiative interaction of the land and the atmosphere. These works have profound significance for the development of meteorological measurement technology today [8]. In addition, Sir Houghton made a significant contribution to the establishment and expansion of the United Nations Intergovernmental Panel on Climate Change (IPCC). Under his guidance, from its inception in 1988 until 2002, the IPCC did an unrivalled job of integrating the science, issuing warnings about dangerous climate risks and making the case for immediate action. The organization won the 2007 Nobel Peace Prize for its efforts. Rainfall is extremely important for the prediction accuracy of propagation delay, and its temporal and spatial variations often have some nonlinear correlations, which is very worthy of attention. In 2022, Varotsos et al. have made some novel studies on the shape of rain and clouds. Based on the concept of “nature rank”, the relationship between cloud area and circumference and rainfall can be obtained by studying the relationship between cloud area and circumference [9]. This is of great significance for accurate meteorological prediction and can also provide certain explanations for the uncertain time-lag relationship between ground water content and precipitation.

In the research field of long-wave propagation delay, it is mainly divided into the study of the calculation model of propagation delay from the terrain and the exploration of the time-varying characteristic and correction method of the ASF value from the time.

The early theoretical calculation of LF groundwave propagation was based on the smooth flat ground channel model [10,11,12,13,14,15,16,17,18]. In 1909, Sommerfeld deduced an expression for the field integral of a vertical electric dipole on a uniform flat surface [10]. Based on this theory, scholars from all over the world have carried out a lot of relevant studies over the years. In 1945, Fock transformed and approximated the spherical harmonic series expression of the spherical groundwave field derived by Watson [11], which improved the convergence rate of the series and was more suitable for engineering applications, namely, the Fock diffraction formula [12]. For smooth segmented uneven ground, the main methods include the Wait integral formula [13], wave-mode conversion method [14] and Millington empirical formula widely used in engineering calculations [15]. For irregular and uneven ground, the evolutionary calculation methods mainly include the parabolic equation (PE) method [16], integral equation method [17] and finite different time domain (FDTD) algorithm [18]. The above methods all have their important theoretical guidance and practical application value in different applicable environments.

In terms of the time-varying characteristics of propagation delay, as early as 1985, McCollough et al. first discovered the seasonal variation of propagation delay through Loran-C observation data in the Gulf of Mexico, and preliminarily discussed its impact [19]. In 1986, Ma made a study on the seasonal variation in ground conductivity in different regions of China and its influence on LF groundwave propagation [20]. In 1991, Ren analyzed the influence of meteorology on the prediction of signal propagation in the Northeast Sea working area of Loran-C [21].

In recent years, with the increasing demand for the timing and positioning accuracy of eLoran systems in various countries, a lot of research work has emerged on the time-varying characteristics of propagation delay and the *ASF* correction method. Yang et al. analyzed the correlation between daily variations in temperature and humidity and propagation delay and proposed a compensation scheme based on temperature for delay correction [22]. Hwang et al. studied the influence of temperature and ground conductivity on the eLoran propagation delay and corrected the *ASF* by establishing a linear temperature model [23]. Li et al. and Yang et al., respectively, compared the measured propagation delay before and after heavy snow and rain and concluded that abrupt changes in meteorological factors such as temperature, humidity and precipitation are the key reasons for drastic changes in *ASF*. [24,25]. Pu et al. and Feng et al. analyzed from the perspective of the atmospheric refractive index and soil moisture content, respectively, and concluded that each factor is closely related to the propagation delay changes [26,27]. Jia et al. obtained a conclusion that the groundwave propagation delay is larger in the day than at night, and the fluctuation increases with distance [28].

In 2019, Yang conducted a preliminary prediction modeling study on the short-term time-varying changes of short distance propagation delay and the fluctuations of various meteorological factors through the neural network principle [29], which broke through the common linear correction model and was an innovative concept but did not consider the regional changes of meteorological factors. In 2020, Wang et al. gridded the long-wave signal region and proposed a correction method based on a neural network (ground conductivity, refractive index of light, temperature and humidity as inputs) [30], which achieved certain effects but took fewer meteorological factors into account.

In summary, the current research on the theoretical calculation methods of LF groundwave signal propagation delay has been relatively mature. However, the research on the time-varying characteristics of propagation delay is still limited, mostly focusing on the analysis of the relationship between weather and propagation delay, and the correction methods are mainly linear models, while the learning algorithms such as neural network models are in their infancy. Aiming at the above deficiencies, this paper makes an in-depth study, and the main contributions of our study are as follows:The influence of meteorological factors on different components of long-wave propagation delay (PF, SF, ASF) is theoretically analyzed quantitatively or qualitatively. Through the correlation analysis of long-term measured data and theoretical analysis, seven relatively important meteorological influence factors are determined as modeling parameters.Considering multiple meteorological factors and their regional changes, the prediction model based on the Back-Propagation neural network (BPNN) is established, which significantly improves the accuracy of propagation delay prediction.The 17 Sustainable Development Goals set out in the UN 2030 Agenda to address major global challenges, including poverty, inequality, climate change, environmental degradation, peace and justice, are set to be achieved by 2030 [31]. The successful realization of the forward prediction of meteorological propagation delay has theoretical reference value for the research on the inversion of meteorological elements, especially the prediction of disaster weather, through long-wave propagation delay in the next project plan and has far-reaching significance for the prevention of poverty, environmental degradation and other challenges.

## 2. Background Principles

Each Loran-C station chain consists of a main station and two to four sub-stations and has a special group repetition interval (GRI). The stations broadcast in cycles according to GRI in a certain time sequence to ensure a fixed receiving sequence within the signal coverage area [32]. Loran-C signals are transmitted in the form of pulse groups, with nine pulses in each pulse group of the main station and eight pulses in each pulse group of each sub-station. The interval of the first eight pulses in each group is 1 ms, while the interval of the ninth and eighth pulses in the main station is 2 ms [33]. The main improvement of the eLoran system on Loran-C is to adopt Eurofix data link technology to perform three state pulse position modulation (PPM) on the third to eighth pulses of each pulse group to achieve the data information transmission function [34,35,36]. The time domain waveform of the standard Loran-C pulse is expressed by Formula (1) [37], and the waveforms of the standard Loran-C pulse and main station pulse group are shown in Figure 1. below.
(1)$$s\left(t\right)=\{\begin{array}{c} A{\left(t-\tau \right)}^{2}\exp\left(-2\frac{\left(t-\tau \right)}{65}\right)\sin\left[2\pi f_{c}t+P_{c}\right],\;\;\tau \ll t\ll 65+\tau \;\\ \;\;\;\;\;\;\;\;\;\;\;\;\;0,\;\;\;\;\;\;\;\;\;\;\;\;\;\;\;\;\;\;\;\;\;t<\tau \end{array}$$
where t is the time in μs; *A* is a normalization constant related to the peak amplitude; *τ* is the envelope-to-cycle difference (ECD) in µs; $f_{c}$ = 100 kHz is the carrier frequency and $P_{c}$ = 0 or *π* is the phase code parameter in radians.

The autonomous timing function of the eLoran system is the process of synchronizing the local 1PPS signal of the receiver with the 1PPS signal of Universal Time Coordinated (UTC). The basic principle of autonomous timing is shown in Figure 2.

According to the time axis relationship shown in Figure 2, the process of autonomous timing is the process of calculating ∆*T*, which can be expressed by the following formula:(2)$$\Delta T\;=\;T_{m}+TOA-T_{d}=T_{m}+T_{p}+T_{r}-T_{d}$$
where *T_m_* is the deviation between the transmit pulse and 1PPS (UTC); *T_p_* is the absolute propagation delay of the eLoran groundwave signal from the transmitting antenna to the receiving antenna; *T_r_* is the delay of the receiving system; *T_d_* is the time deviation between the receiver’s local 1PPS time and the Group Trigger Pulse (GTP) time [38].

The GTP time corresponds to the standard zero crossing of the first pulse signal of the eLoran pulse group. Through the period identification process, the eLoran receiver determines the GTP time and tracks the phase of the standard zero-crossing point; then, it accurately measures the TOA of the groundwave signal and finally realizes the timing and position service functions. According to Formula (2), TOA is composed of two parts of *T_p_* and *T_r_* [39], where *T_r_* can be obtained by calibration. In addition, *T_m_* in the formula can be calculated by demodulating the time-code information, and *T_d_* can also be measured by the internal counter of the receiver. Therefore, it can be said that the core of calculating ΔT and realizing accurate autonomous timing is the accurate estimation of propagation delay *T_p_*. The expression of *T_p_* is as follows [39,40]:(3)$$T_{p}\;=\;\mathrm{PF}+\mathrm{SF}+\mathrm{ASF}$$
where PF is the primary factor; SF is the secondary factor; ASF is the additional secondary factor. The above parameters all have their characteristics and estimation methods, which will be discussed in Section 3. Among them, since the calculation of *T_p_* components only considers the topographic factors, the *T_p_* measurement results of the fixed broadcasting stations at the same test site should be a fluctuation with a horizontal trend, taking into account the influence of some random noise or sudden jump interference.

However, through the analysis of long-term fixed-point observation data, it is found that the *T_p_* values present a real-time fluctuation trend. Figure 3 displays the curve of *T_p_* data collected at the Lintong BPL long-wave monitoring station within the 184-day interval (1 July to 31 December 2021) after the processing of outlier removal and mean value removal. It should be noted that all the *T_p_* data used in this paper are derived from eLoran groundwave signals broadcast by a BPL long-wave timing station.

As can be seen from the *T_p_* curve in Figure 3, the measured values collected at the Lintong monitoring Station (71.4 km) have a slow seasonal change process of nearly 150 ns in half a year, and there also seems to be some regular fluctuation in the diurnal trend. However, the measurement results of the antenna bottom current loop obtained from the monitoring station in Pucheng show that the transmitting second signal is in a long-term stable state, the maximum jitter deviation fluctuates around ±20 ns and the standard deviation is less than 11 ns. Therefore, it can be preliminarily judged that the measured value fluctuation of propagation delay has a certain correlation with the change in propagation medium characteristics on the fixed propagation path. In this paper, parameter ∆*T_p_* will be used to characterize the total relative fluctuation of such propagation delay *T_p_* and as the core research object of this paper.

## 3. Correlation Analysis

The navigation and positioning functions of the eLoran system are based on the *T_p_* measurement of groundwave signals. Many previous studies have found that, on the relatively fixed groundwave propagation path, the main factors affecting the trend of ∆*T_p_* are various meteorological changes, including temperature, relative humidity, atmospheric pressure, etc., which may disturb the characteristics of the propagation medium [19,20,21,22,23,24,25,26,27,28,29,30]. Indirectly, the *T_p_* value also changes.

The following will discuss the theoretical influence of meteorological fluctuations on PF, SF and ASF, which will be demonstrated separately in the following sections, and the complex correlations between various meteorological factors and ∆*T_p_* will be demonstrated through sufficient measured data.

### 3.1. Theoretical Influence of Meteorological Factors

#### 3.1.1. PF

The primary factor, also known as the basic time delay, refers to the signal propagation delay under the assumption of a uniform air medium, which can be calculated by Formula (4). In order to reflect the spatial difference of the upper atmospheric refraction index in more detail, including the influence of altitude, temperature and humidity differences in different regions, and improve the estimation accuracy, Formula (4) can also be changed to Formula (5).
(4)$$\mathrm{PF}=\frac{n_{0}s}{c}$$
(5)$$\mathrm{PF}=\int \frac{n_{s}}{c}ds$$
where *s* is the great-circle distance of the propagation path in km; *c* is the speed of light in vacuum, which is 299,792.458 km/s; *n*_0_ is the standard atmospheric refraction index, which takes *n*_0_ = 1.000315 in China’s electromagnetic industry standard; *n*_s_ is the piecewise atmospheric refractive index related to the spatial inconsistency of the propagation path.

It can be seen from the above introduction that for fixed eLoran signal receiving points, the theoretical calculated value of PF should be a definite value. However, considering that the atmospheric refraction index *n* is a time-varying parameter in the real environment, the real-time fluctuation and compensation of PF must be a problem worth studying. In order to facilitate the analysis, another parameter, *N* (unit: N), is commonly quoted to replace the atmospheric refractive index *n*, and it can be expressed as
(6)$$N=\left(n-1\right)\cdot 10^{6}$$

According to the international standard atmosphere, the standard value of parameter *N*_0_ = 315 (*n*_0_ = 1.000315) is adopted in the electromagnetic industry in China, while *N*_0_ = 338 (*n*_0_ = 1.000338) is adopted in the United States. This value deviation also reflects the great differences in geological types and climate characteristics between different countries. Considering that the relative permeability in the air is approximately equal to 1, the relative dielectric constant $\varepsilon_{A}$, electric polarization $\chi_{e}$ and parameter *N* of the atmosphere are related as follows [41]:(7)$$N\approx \frac{\left(n^{2}-1\right)}{2}\cdot 10^{6}$$
(8)$$\chi_{e}=\varepsilon_{A}-1=\frac{155.2\times 10^{-6}\cdot P}{T}+\frac{7.645\times 10^{-1}e_{w}}{T^{2}}$$
where *P* is the atmospheric pressure in hPa; $e_{w}$ is the water vapor pressure in hPa; *T* is the temperature in K. In addition, when the frequency of radio waves is below the UHF band, especially in the LF band, the transformation relation of $\varepsilon_{A}=n^{2}$ can be established [42]. By combining this equation with Equations (7) and (8), the approximate calculation formula of *N* can be obtained [43]:(9)$$N\approx 77.6\frac{P}{T}+3.73\times 10^{5}\frac{e_{w}}{T^{2}}$$

Table 1 shows the seasonal variation in parameter *N* in different regions of China [41]. The statistics show that the distribution of parameter *N* varies greatly in different regions, and its value fluctuation is also very different; especially, it is extremely sensitive to humidity changes. Due to the strong seasonal variation in humidity and temperature in the coastal and middle reaches of the Yangtze River, the value of *N* fluctuates obviously. However, Xinjiang, Xizang and other regions have relatively stable fluctuations of *N* due to the dry climate, less precipitation and little change in atmospheric humidity.

At the same time, the research shows that the diurnal variation in parameter *N* also has obvious regularity. Generally, in the early and late hours of every day, the temperature is low and the humidity is high; in the afternoon, as the temperature rises, the humidity decreases accordingly. Therefore, the value of *N* is larger in the morning and evening and reaches its minimum around 14 pm. Generally speaking, the diurnal variation range of *N* in most regions is 10~20 N under the normal weather environment [44]. According to Formula (4), it can be calculated that for parameter N, the calculation error of PF brought by the fluctuation of 1 N is 3.3 ps/km. Therefore, at 1000 km, the PF term may fluctuate by about 67 ns due to a variation of 20 N, while in the case of meteorological anomalies, the influence caused by the short and drastic variation in *N* may even reach 200 ns.

#### 3.1.2. SF and ASF

SF refers to the additional influence of the seawater medium on signal propagation delay. ASF refers to the additional influence of the difference in conduction characteristics of signals in land and seawater media on the propagation delay, which is the disturbance compensation of different media based on the secondary time delay. The two parameters are essentially the same, that is, to calculate the groundwave attenuation function. The logical expression of SF + ASF can be described here as follows:(10)$$\mathrm{SF}\;+\mathrm{ASF}\;=\;\frac{10^{6}}{\omega }\arg(W)$$
where ω is the central angular frequency of the signal, and W is the groundwave attenuation function, which varies with distance s and is also the core of delay measurement accuracy. When the signal propagation distance *s* is relatively short, *W* is generally calculated with a flat ground model, while when *s* > 70 km, the influence of the spherical curvature of the ground on signal propagation needs to be considered [5,45]. The calculation formula of *W* is very complex, which is related to coefficients such as *d* and normalized surface impedance *q* and corresponds to different expressions, respectively. For details, refer to the literature [46,47], where only the calculation formula of *q* is given directly:(11)$$q=\text{-}i{\left(\frac{K_{1}a_{e}}{2}\right)}^{\frac{1}{3}}\frac{n\sqrt{{\varepsilon \text{-}n}^{2}\text{-}i60\lambda \sigma }}{\varepsilon \text{-}i60\lambda \sigma }$$
where ε is the ground relative dielectric constant of the propagation path; σ is the ground conductivity; $K_{1}$ is the wave number in the air; $a_{e}$ is the equivalent earth coefficient; λ is the wavelength in vacuum.

According to Formula (11) and the description in reference [47], except for the fixed parameters, the normalized surface impedance *q* is the function of ε, σ and *n*, in which the influence of *n* is small, and the constant 1 is generally taken in calculations. Especially for eLoran signals in the LF band, the ground conductivity is the dominant factor affecting the propagation characteristics of the groundwave. Therefore, in the theoretical analysis, the attenuation function *W* in Formula (10) can also be recorded as W(σ, ε, s).

Different geological types have different effects on the absorption, refraction and reflection of signal propagation. Therefore, the finer the regional division of ground conductivity and the relative dielectric constant is, the more accurate the measurement of SF (ASF) will be. At present, there are several works of research on electronic digital maps of ground conductivities [48,49], which are crucial to the refinement correction of *T_p_*, but they do not fully consider the real-time changes in σ and ε.

The factors determining the value of σ and ε mainly include: (1) geological structure; (2) ground vegetation; (3) underground layered structure; (4) electromagnetic wave frequency. The latter three items are generally stable, while the conductivity of the geological structure fluctuates relatively easily due to the soil moisture and salinity caused by real-time environmental changes such as precipitation and temperature fluctuation. The research conclusion of the literature [50] points out that there is a strong positive correlation between the soil water content and dielectric constant. Seasonal changes in climate and diurnal temperature variation will cause the change in soil moisture. Especially in one day, due to the low temperature in the morning and evening, there is more water in the soil, and the value of ε is larger. At noon, as the temperature rises and the water vapor evaporates, the soil will be slightly dry, and the value of ε will decrease accordingly.

Table 2 shows the conductivity characteristics of nine typical ground types, respectively. It can be seen that σ and ε of different geological characteristics are very different, and their time-varying characteristics will also affect the selection of equivalent earth radius parameters in the calculation formula, which will inevitably bring calculation errors and together lead to drastic changes in the calculated values of ASF.

The attenuation function *W* can be calculated using the integral equation method, and the SF (ASF) change curve can also be obtained, as shown in Figure 4. Changes in SF + ASF under different ground types clearly show the huge differences in the attenuation function estimation caused by the characteristics of different conductive media. The numerical difference between the two points A and B indicates that the difference in ASF values between the two different ground types of average land and wet ground is more than 1 μs over the propagation path of 400 km. In the real environment, heavy precipitation and other reasons are also the main reasons for real-time changes in ground conductivity and the occurrence of such delay deviations, which is unlikely to be corrected by the real-time measurement of ground conductivity. It can be said that the fluctuation of ASF is also the most important component of the overall ∆*T_p_*.

### 3.2. Anaylsis of Measured Data

There are many types of meteorological factors, which may have different effects on signal propagation. Therefore, it is necessary to first understand whether a certain parameter has an impact on *T_p_* and the magnitude of the impact. In the multivariate analysis of vectors of different types, a correlation coefficient is generally needed. The three commonly used correlation coefficients in statistics are the Pearson, Kendall and Spearman correlation coefficients [51], all of which reflect the consistency and similarity of variation trends among different variables and have the same value range (−1~1) but also differ in application scopes and characteristics.

The Pearson correlation coefficient (PCC) is a method for measuring the degree of linear correlation between variables. It is defined as the quotient of the covariance and standard deviation of the two variables *X* and *Y*, which is often expressed by the Greek letter $\rho_{XY}$. In large-sample statistics, the expected value can be replaced by the sample mean according to the sample number *m*, and Formula (12) can be used for approximate calculation.

The principle of the Spearman correlation coefficient is to use the rank of two variables for linear correlation analysis, usually expressed by the Greek letter $\rho_{s}$. When calculating $\rho_{s}$, it is necessary to sort the original data groups respectively by size to obtain the corresponding rank difference $d_{i}$, and then calculate according to Formula (13).

Similar to $\rho_{s}$, the Kendall correlation coefficient also depends on rank and has the same requirements on data conditions. Generally, it is expressed by the Greek letter τ, which is used to reflect the correlation of classification variables. According to the number of harmonious and disharmonious pairs *C* and *D* of data groups X and Y, τ can be defined as the ratio of the difference between the two numbers and the total pairs as Formula (14).
(12)$$\rho_{XY}=\frac{m\sum_{i=1}^{m}X_{i}Y_{i}-\sum_{i=1}^{m}X_{i}\sum_{i=1}^{m}Y_{i}}{\sqrt{m\sum_{i=1}^{m}X_{i}{}^{2}-{\left(\sum_{i=1}^{m}X_{i}\right)}^{2}}\sqrt{m\sum_{i=1}^{m}Y_{i}{}^{2}-{\left(\sum_{i=1}^{m}Y_{i}\right)}^{2}}}$$
(13)$$\rho_{s}=1-\frac{6\sum_{i=1}^{m}d_{i}{}^{2}}{m\left(m^{2}-1\right)}$$
(14)$$\tau =\frac{C-D}{\sqrt{\left(\frac{1}{2}m\left(m-1\right)-\frac{1}{2}\sum_{u=1}^{s}U_{u}\left(U_{u}-1\right)\right)\left(\frac{1}{2}m\left(m-1\right)-\frac{1}{2}\sum_{v=1}^{t}V_{v}\left(V_{v}-1\right)\right)}}$$

Among the three types of analysis methods, PCC measures the linear relationship between variables and is very sensitive to extreme outliers. The Spearman and Kendall correlation coefficients, as nonparametric methods, are both based on the rank and relative size of the observed value. They have a stronger tolerance to outliers, mainly measure the relationship between variables and can describe nonlinear correlation to a certain extent; therefore, they are more widely applicable than PCC. Due to the complex relationship between the measured *T_p_* and various meteorological factors, it cannot fully meet the applicable conditions of any of the above methods, and only using a certain correlation analysis method may lead to the error of ignoring the impact factors due to the possible low correlation coefficient. Therefore, in our study, the three correlation analysis methods were simultaneously used to conduct the joint analysis and mutual verification of the long-term and short-term correlation between meteorological factors and propagation delay so as to find out more meteorological factors causing ∆*T_p_* as much as possible.

#### 3.2.1. Seasonal Correlation

According to the analysis in Section 3.1, changes in meteorological factors such as temperature, relative humidity and atmospheric pressure will affect the calculation factors of *T_p_* and indirectly lead to its fluctuation, which is verified by the measured data below.

Figure 5a shows the curve of the long-term monitoring *T_p_* data in Figure 3 after pre-processing such as smooth filtering, outlier removal, down-sampling and mean removal. Meanwhile, Figure 5b shows the curve of meteorological data collected by the meteorological station (Weinan) on the corresponding signal propagation path in the same period.

It can be seen intuitively from Figure 5 that the seasonal change in *T_p_* has a strong linear correlation with the temperature and atmospheric pressure. The statistical data under different correlation analysis methods in Table 3 also verify this conclusion. The PCC values of temperature and atmospheric pressure can reach 0.8 and −0.7 respectively. It can be seen that within a propagation distance of about 80 km, the temperature difference of more than 40 °C leads to the peak-to-peak ∆*T_p_* exceeding 150 ns. This result also verifies the theoretical analysis in Section 3.1 regarding the disturbance of meteorological factors in the calculation of *T_p_* and the conclusions of previous relevant studies.

#### 3.2.2. Diurnal Correlation

According to the analysis results of seasonal correlation, it is not difficult to speculate that the short-term diurnal variation in *T_p_* should also be very strong because the temperature difference, rainfall, snowfall and other meteorological factors will also fluctuate dramatically within a day. Figure 6 and Figure 7 show the curves of short-term *T_p_* data collected at different test sites within 10 days and the meteorological data curves of the corresponding meteorological station, including Jingyang (9 January 2021~18 January 2021, 70 km) and Meixian (30 September 2018~9 October 2018, 184 km). Among them, the *T_p_* data in the figure were pre-processed in the same way as in Figure 5a.

The curve in the figures indicated that the ∆*T_p_* of different test locations vary dramatically within a day. In Meixian, where the propagation path is relatively long, the fluctuation of some days even exceeds 200 ns. In addition, the daily variation trend of Tp is similar to a certain extent, generally reaching the peak value around 14:00 to 15:00 and reaching the minimum value in the early morning, which has a strong consistency with the variation trend of temperature and relative humidity.

Table 4 shows the correlation coefficient between the daily measured *T_p_* and the three factors of temperature, relative humidity and atmospheric pressure, respectively. Due to the large amount of data, only the PCC values are listed in the tables, while the other two types of unlisted correlation values have little difference. Among them, the linear correlation between temperature and *T_p_* is extremely high. Due to the close distance between Jingyang and the BPL station and the little change in the propagation path, the average PCC of temperature even exceeds 0.91, while the PCC of humidity reaches −0.89, and the correlation of atmospheric pressure also reaches a very high level.

At the same time, it can be seen that the PCC values of temperature and humidity decreased significantly at the far away test site in Meixian. On the one hand, on the first, seventh and eighth days, the receiver or the antenna failed, respectively, resulting in the measurement error of *T_p_* and the abnormal value of the correlation coefficient, which also affected the mean value. On the other hand, due to the longer propagation path, the meteorological changes in other parts of the path and other unconsidered meteorological factors may also affect the signal propagation characteristics, resulting in a lower correlation coefficient when the equipment is normal.

In order to identify more meteorological factors that may cause the ∆*T_p_*, while fully considering their regional changes, this paper applied different correlation analysis methods to jointly analyze the 60-day (January to March, 2022) continuous *T_p_* data collected at the Jingyang test set and different meteorological factors from multiple meteorological observation stations on the same propagation path. Then, seven main meteorological factors including temperature, relative humidity, atmospheric pressure, water vapor pressure, wind speed, wind direction and precipitation with a relatively strong correlation were identified. The correlation coefficient statistics are shown in Table 5.

As can be seen from the data in Table 5, there are obvious differences in the degree of correlation between different meteorological factors, and the same factor also has certain regional differences. For different correlation analysis types, the correlation characteristics of each meteorological factor are consistent, which also achieves the effect of mutual verification. In general, temperature and relative humidity have the greatest impact on *T_p_*, but other meteorological factors in the table also have non-negligible influence.

### 3.3. Correlation Discussion

The above theoretical analysis indicates that the real-time mutation of the three parameters of atmospheric refractive index n, ground conductivity σ and relative dielectric constant ε will cause the calculation deviation of PF, SF and ASF, which is also the quantitative reflection of the influence of time-varying meteorological factors on *T_p_*. The maximum fluctuation range of ∆*T_p_* even reaches the microsecond level.

First, the fluctuation of PF is mainly affected by the time-varying characteristics of the atmospheric refractive index n, and the maximum fluctuation range within the coverage of the eLoran signal is generally in the order of 100 ns. At present, there have been some compensation models based on temperature, pressure factors [52], which can partially correct some delay fluctuation errors. However, the applicable environment is mainly the sea area, and the impact of ASF fluctuation caused by changes in a ground conductive environment is not fully considered.

Second, the fluctuation of SF and ASF is mainly affected by the time-varying characteristics of σ and ε, and the maximum variation range may reach the order of microseconds, which is the core component of ∆*T_p_*. From the spatial dimension, the accuracy of calculation depends on the precision of the terrain estimation and geodetic conductivity map. However, from the time dimension, due to the extremely complex correlation between meteorological disturbance and ground characteristics, there is no accurate compensation method for the fluctuation error.

Moreover, according to sufficient measurement data and different correlation coefficient analysis methods, it can be found that seven meteorological factors mentioned in Section 3.2.2 are closely related to the real-time ∆*T_p_* in different degrees. Among them:
Temperature and relative humidity have a strong linear correlation with *T_p_*, and are the leading factors affecting the ∆*T_p_*.The correlation of atmospheric pressure and vapor pressure is relatively stable, which will indirectly lead to ∆*T_p_* fluctuation by affecting the characteristics of the propagation medium, but the linearity is limited.Although the linear correlation is not obvious, the wind direction, wind speed and precipitation may have a complex hidden correlation with the propagation medium. For example, wind speed and direction will affect the visibility, temperature and humidity of the atmosphere and also seriously affect the roughness of the sea surface [53,54]. On the other hand, rain and snow can change the ground conductive properties in non-real time [55].In addition to the unpredictable and serious impact of the propagation medium changes analyzed above on the propagation characteristics of electromagnetic waves, the regional differences in meteorological factors on the propagation path are also issues that must be considered in the study of propagation delay correction.

The nominal timing accuracy of the eLoran system is 1 μs, and the future realization of differential technology requires that the timing accuracy be optimized to within 100 ns in the differential region. The compensation method of ∆*T_p_* caused by meteorological disturbance is one of the urgent problems to be solved. According to the above analysis, a variety of meteorological factors have complex correlations with long-term seasonal and short-term daily changes in *T_p_* that cannot be ignored, which may not be well described by simple models. Therefore, the nonlinear fitting ability of a BP neural network algorithm is considered in this paper to explore the complex functional relationship between ∆*T_p_* and multiple meteorological factors and establish a prediction model of the overall fluctuation of *T_p_* so as to achieve the accurate correction and prediction function of *T_p_*.

## 4. Propagation Delay Prediction Model

### 4.1. Linear Regression Model

Previous studies mainly compensate for ∆*T_p_* by establishing a traditional linear model. Especially for the Loran-C system applied at sea, the linear correlation between temperature and *T_p_* is very strong, so the linear temperature model also has stable fitting and prediction capabilities. The mathematical form of the model is shown in Formula (15).
(15)$$\hat{y}=Kx+B$$

According to the residual between the fitting and the measured value $\hat{y}-y$, the mean square error $E_{liner}$ of the sample is obtained, and then the partial derivatives of the two coefficients K and B of the linear model are calculated as
(16)$$\frac{\partial E_{liner}}{\partial K}=\frac{\partial \sum_{i=1}^{m}\frac{1}{m}{\left({\hat{y}}^{i}-y^{i}\right)}^{2}}{\partial K}=\frac{2}{m}\overset{m}{\underset{i=1}{\sum }}\left[x^{i}\left(Kx^{i}+B-y^{i}\right)\right]$$
(17)$$\frac{\partial E_{liner}}{\partial B}=\frac{\partial \sum_{i=1}^{m}\frac{1}{m}{\left({\hat{y}}^{i}-y^{i}\right)}^{2}}{\partial B}=\frac{2}{m}\overset{m}{\underset{i=1}{\sum }}\left(Kx^{i}+B-y^{i}\right)$$

Based on the principle of least squares (LS), Formulas (16) and (17) are set to zero and solved simultaneously, and the following results can be obtained:(18)$$K=\frac{m\sum_{i=1}^{m}x_{i}y_{i}-\sum_{i=1}^{m}x_{i}\sum_{i=1}^{m}y_{i}}{m\sum_{i=1}^{m}x_{i}{}^{2}-{\left(\sum_{i=1}^{m}x_{i}\right)}^{2}}$$
(19)$$B=\frac{\sum_{i=1}^{m}x_{i}^{2}\sum_{i=1}^{m}y_{i}^{2}-\sum_{i=1}^{m}x_{i}\sum_{i=1}^{m}y_{i}}{m\sum_{i=1}^{m}x_{i}^{2}-{\left(\sum_{i=1}^{m}x_{i}\right)}^{2}}$$

For a terrestrial environment, the propagation characteristics of the LF signal are more complex, and the effect of the linear model may be somewhat reduced. However, considering the high linear correlation of temperature in the above analysis, a temperature-based linear prediction model of *T_p_* is also established in our study, and its performance with other models is compared and analyzed in the following text.

### 4.2. BPNN Model

Machine learning can be understood as the task assigned by a computer, which uses its high-speed computing power to select customized rules for learning according to limited data samples so as to obtain human-like intelligence, constantly optimize the mathematical model, form a complete program and achieve the predetermined task objectives. Traditional machine learning regression models include the support vector machine (SVR), artificial neural network model (ANN), random forest (RF), etc. [56], which are commonly used for material fatigue damage, residual life and other performance prediction methods. The applicability of several types of methods varies [57]. Among them:

SVR is a learning method based on regression analysis, which has good generalization ability and robustness and can obtain good results for small sample problems. SVR are widely used on datasets that are not too large and have obvious relationships.

The RF model is an integrated learning algorithm for regression problems composed of sets of different classification or regression trees. During the training, the algorithm selects samples randomly several times, establishes the decision tree model and returns the average value of multiple models or most of the results as the final prediction result. The construction of robust quantitative decision trees is the key to the prediction accuracy of RF models. RF has a good effect when processing large datasets at a high latitude.

ANN is a highly complex nonlinear model composed of many neurons. In this system, individual neurons only have relatively simple application functions and type construction. However, when combined into an entire system, their behavior can be very complex, and they are often used for highly nonlinear problems. One of the outstanding advantages of this algorithm is that it has strong self-learning and adaptive ability.

Each of the above methods has its own advantages and has been widely used in many fields such as medicine, architecture and data processing. In this study, because there may be various nonlinear relationships between various meteorological factors and eLoran propagation delay and moderate data scales, we chose the BP neural network, which is more suitable for a strong nonlinear relationship, for modeling. At the same time, it can also verify the performance improvement of this research under the same background through the repetition of previous achievements. In the later deeper study (including high-density and multi-type meteorological data collection), we will also consider using more prediction models including the RF model for comparison and optimization studies.

ANN is a mathematical calculation model based on the structure and working principle of a biological neural network. Its basic constituent structure is the neuron, as shown in Figure 8a. Its mathematical composition is very simple, including the weight vector ω, threshold value (bias) θ and activation function f. The activation function is to carry out the nonlinear transformation of the calculated results so as to improve the mapping ability of the neural network to deal with nonlinear fitting work. Figure 8 shows three activation functions commonly used in BP neural networks. After each input vector x enters the neuron, it outputs the result in the form of $y=f\left(\omega^{T}x-\theta \right)$.

BPNN is the most successful and widely used feedforward neural network learning algorithm derived from the cross and fusion of multiple neurons. Its main feature is that the signal propagates forward, while the error propagates backward [58]. The weight value of the network coefficient is corrected layer by layer through adaptive learning in the inverse propagation direction so as to reduce the output error of the network as much as possible, finally achieving the purpose of training and realizing the nonlinear fitting and prediction function. The principle of the single-hidden layer BPNN model can be described in the form in Figure 9b, and the multi-hidden layer is similar [59].

According to the analysis in Section 3, multiple meteorological factors have direct or indirect effects on the real-time changes in *n*, σ, ε and other factors that may affect the propagation characteristics of electromagnetic waves, and the correlation is extremely complex to be described by simple mathematical models. Relying on its powerful nonlinear mapping, fault tolerance and generalization capabilities, this paper considers skipping the indirect correlation and uses BPNN to establish direct mapping between meteorological factors and *T_p_* so as to improve the real-time prediction accuracy of ∆*T_p_*.

For the problems discussed in this paper, the number of hidden layers can be 1~3 according to different input conditions. As shown in Figure 9b, the parameter derivation and modeling of BPNN with a single-hidden layer structure are carried out as follows:
Network Initialization. The cellular samples composed of various meteorological data and ∆*T_p_* should be divided into a training set and test set in proportion (generally, 7:3~9:1) and normalized to prevent overflow during training. Here, the training set D={(x1,y1),(xk,yk),⋯,(xn,yn)}, xk∈ℝd,yk∈ℝl. According to the empirical Formula (20), the range of the number of hidden layer nodes can be estimated, and the best value q can be determined by the trial-and-error method. In addition, set the learning rate and activation function and initialize the connection weights and activation function thresholds between layers.
(20)p=d+l+M,  1≤M≤10Input→Hidden→Output. In the network shown in Figure 8b, three layers contain *d*, *p* and *l* neurons, respectively, and the forms of connection weights of forward transmission between layers are νih and ωhj respectively. The forms of activation function thresholds of the hidden and output layer are γh and θj respectively, and then the corresponding activation function outputs can be expressed as:(21)bh=f(αh−γh)=f(∑i=1dνihxi−γh)
(22)y^j=f(βj−θj)=f(∑h=1pωhjbh−θj)Iterative Loss Function. For the *k*-th training example (xk, yk), after the activation function, the output layer obtains yk=(y1k,y2k,⋯,ylk). Then, the loss function *E_k_* can be obtained.
(23)Ek=12∑j=1l(yjk−y^jk)2Reverse Gradient Calculation. The BPNN algorithm is based on the Stochastic Gradient Descent (SGD) strategy and updates parameters in the negative gradient direction of the target. For the iterative error *E_k_*, set the learning rate ηω, ην, calculate the partial derivatives of the target parameters of each layer in reverse order and then the corresponding gradient update value can be obtained as follows:(24)Δωhj=−ηω∂Ek∂y^jk·∂y^jk∂βj·∂βj∂ωhj=ηω·y^jk(1−y^jk)(yjk−y^jk)·bh=ηωgjbh
(25)Δθj=−ηω∂Ek∂θj=−ηω∂Ek∂y^jk·∂y^jk∂βj·∂βj∂θj=−ηωgj
(26)Δνih=−ην∑j=1l∂Ek∂y^jk·∂y^jk∂βj·∂βj∂bh·∂bh∂αh·∂αh∂νih=ην∑j=1lgj·ωhj·bh(1−bh)·xi=ηνehxi
(27)Δγh=−ηv∂Ek∂γh=−ηv∑j=1l∂Ek∂y^jk·∂y^jk∂βj·∂βj∂bh·∂bh∂αh·∂αh∂γh=−ηvehThe above derivation process uses the feature of the sigmoid function: f′(x)=f(x)f(1−x) and resolves the complex partial derivative operation of each parameter into a simple form, which greatly facilitates the calculation (similar to other activation functions).Parameter Update and Iteration Judgment. Judge whether the training process meets the termination conditions. If not, adjust the parameters according to the calculation result of the updated values and return to step 2 for the next training.

The above derivation process illustrates the basic framework and execution process of the BPNN model. Based on this, in view of the environment of multiple meteorological factors in multiple regions, different BP propagation delay prediction models are established in this study, and the optimal model is obtained by adjusting the input meteorological factor type, hidden layer structure, learning rate, gradient overflow range and other parameters for model modification.

The convergence process of BPNN will be significantly affected by the learning rate. Too large of a learning rate will lead to overfitting, while too small of a learning rate may not converge [57]. Considering the similarity between the BPNN and LMS filter, we choose to use the grid search method and select a traversal range and step size (e.g., 0.01~0.5, 0.01) of the linear ergodic parameter adjustment. The specific parameters and performance comparisons of all models are analyzed in detail in Section 5.

## 5. Performance Analysis and Discussion

This section tests and compares different BPNN models and linear models under a variety of conditions, and the optimal model parameter selection will be uniformly displayed in Section 5.4.

### 5.1. Data Acquisition and Processing

The data used to build the prediction model include the long-term continuous measurements of *T_p_* collected from the Jingyang test site and the statistical values of different meteorological factors at each meteorological station along the corresponding signal propagation path during the same period. Among them:
Propagation delay data
Broadcasting station: BPL long-wave timing station in Pucheng;Acquisition equipment: eLoran timing receiver (KTL-101B), long-wave receiving electric antenna (KTL-606A), GPS receiver, GPS receiving antenna, counter (SR620);Collection site: Longquan Commune, Jingyang County, Xianyang City;Collection period: 60 days (from December 2021 to March 2022, excluding some days of power failure and equipment failure);Sampling rate: one time per second.

After the same preprocessing process as described in Section 3.2.1, such as smooth filtering, outlier removal, down-sampling and mean value removal, the collected data can maintain the time corresponding relationship with the meteorological data and clearly show its fluctuations. The processed data are shown in Figure 10.
b.Meteorological data
Collection sites: four national meteorological observation stations in Pucheng, Fuping, Sanyuan and Jingyang;Collection period: corresponding to propagation delay;Data types: temperature, relative humidity, air pressure, water vapor pressure, wind speed, wind direction, amount of precipitation;Data resolution: one time per hour.

The red dots in Figure 11 mark the locations of national meteorological stations, four of which are adjacent to the propagation path (blue line). Due to the long collection period, during which rain and snow or drastic temperature changes were experienced, the overall data of the measured *T_p_* are representative to a certain extent.

### 5.2. Single-Factor Model

The correlation analysis results in Section 3.2.2 show that temperature and relative humidity have a strong linear correlation with *T_p_*, with PCC values up to 0.91 and −0.89, respectively. Therefore, in this section, temperature and relative humidity are used as network inputs to adjust neural network parameters and train to obtain a single-factor BPNN prediction model, establishing the linear model for analysis and comparison.

First, 54 days (90%) of observation data were randomly selected to build the training set, and the other 6 days (10%) of data were used as the test set. Secondly, the temperature and relative humidity data of the Jingyang meteorological station were used as the single-factor input, respectively, the number of hidden layers was selected as 1~2 and the number of hidden layer nodes was selected as 3~12 to build different BPNN models. Then, according to the principles of minimum root mean square error (RMSE), the optimal models were obtained, which are called the Jingyang time-delay temperature (single point) BP neural network (BP-JTT1) and Jingyang time-delay relative humidity (single point) BP neural network (BP-JTH1), respectively. The test results are shown in Figure 12.

According to the effect comparison displayed in Figure 12, the model BP-JTT1 established based on the temperature factor has a higher fitting degree than BP-JTH1 based on relative humidity. The RMSE and mean absolute error (MAE) of BP-JTT1 are 9.4467 ns and 10.2120 ns, while those of the BP-JTH1 model are 12.7932 ns and 13.1453 ns.

Both of the models have stable fitting and prediction capabilities for propagation delay fluctuations, but because of the stronger linear correlation of temperature factors, the prediction effect is also significantly better. Therefore, a linear prediction model based on temperature was established for comparative analysis. By substituting the same training set data above into Formulas (18) and (19), the linear model coefficients *K* = 2.4065 and *B* = 3.3738 can be obtained. Use the test set data to verify the model, and the effect comparison is shown in Figure 13.

Data statistics show that the RMSE and MAE of the linear model are 9.2591 ns and 11.2292 ns, and the maximum absolute error (MAXE) is 35.9205 ns. Several indexes are all slightly inferior to those of the BP-JTT1 model, which also verifies the results and conclusions of previous studies on the single-factor BP model and linear model [22,23,29]. Especially, when the weather changes abruptly and the temperature is relatively stable, the meteorological disturbance factors may affect the linearity of the temperature model, so the uncertainty of the model will increase and the fitting effect will further decline. This is also the inevitable result of a simple model and too few interference factors taken into account.

### 5.3. Multi-Factor Model

The single-factor BPNN model BP-JTT1, BP-JTH1 and the linear model introduced in the previous section have stable fitting and prediction effects, but the accuracy is general. Therefore, this section took into account the influence of regional changes and the diversity of meteorological factors to propose the multi-dimensional models.

First, the influence of multiple meteorological regions was considered. According to the temperature statistics of four meteorological stations along the propagation path from Pucheng to Jingyang, a variety of BP neural network models with 1~2 hidden layers and 4~13 hidden layer nodes were established, and the optimal model BP-JTT4 was obtained. According to statistics, the number of hidden layers and nodes of this kind of BP model, which only considers temperature factors, has little influence on the fitting effect. However, there may be training overflow errors in a single-hidden layer structure. Therefore, it is recommended to design the hidden layer as a two-layer structure.

The prediction effect of the optimal BP-JTT4 model is shown in Figure 14. According to statistics, the RMSE, MAE and MAXE indexes of the BP-JTT4 model are 8.5696 ns, 6.4151 ns and 23.8801 ns, respectively. It can be seen that when considering the joint influence of multiple related meteorological regions at the same time, the effect of the BP temperature prediction model is improved to a certain extent compared with the BP-JTT1 model, which can cope with the temperature changes in different regions on a longer propagation path.

Although the BP-JTT4 model has been able to achieve good fitting and prediction results, the impact of more related meteorological factors, especially the recessive or weak correlation factors, cannot be quantified.

In view of the diversity of relevant meteorological factors, we also proposed the multi-layer BPNN models with an input layer dimension of 28 on the basis of BP-JTT4 to fit the fluctuation of propagation delay. The data of the network input layer include seven related meteorological factors such as temperature, relative humidity, atmospheric pressure, water vapor pressure, wind speed, wind direction and rainfall collected from four meteorological observation points along the signal transmission path. The number of hidden layers selected is 1~3, and the number of nodes selected is 7~16, 4~13 and 4~13, respectively. Then, according to the minimum RMSE principle, the optimal prediction model, called BP-JTM4, was obtained, and its effect is shown in Figure 15.

Obviously, when considering a variety of meteorological disturbance factors and their regional changes, the prediction effect of ∆*T_p_* is excellent. The coincidence degree between the predicted curve and the measured curve is quite high, and the performance index also reaches the best value. The RMSE and MAE indexes reach 6.2457 ns and 5.0817 ns, respectively, and the MAXE value is only 14.8372 ns.

### 5.4. Comprehensive Performance Analysis

For accurately predicting ∆*T_p_*, the above contents in this section studied and established the BPNN prediction models with various input conditions and types of meteorological factors and the temperature linear prediction model. Meanwhile, a statistical analysis and a comparison of the predicted values of each model with the same test set were conducted. The performance indexes of each model and the optimal structure composition of the four types of BPNN models are respectively shown in Table 6 and Table 7.

Through the above modeling analysis and performance statistics, the following conclusions can be drawn:The three models with a single input factor all have a certain fitting ability, among which the temperature model performs better than the relative humidity model and the linear model, which verifies the conclusions of previous studies. The setting of a single hidden layer is sufficient to achieve the optimal state of the single-factor BPNN model.BPNN can effectively deal with the problem of the ∆*T_p_* prediction of the eLoran groundwave, and with the increase in the input layer correlation factor dimension, the adaptive training ability of the multi-layer network is fully reflected. Compared with the BP-JTT1 model, the fitting stability and accuracy of the BP-JTM4 model are significantly improved, and its RMSE and MAE decreased by 33.88% and 50.24%, respectively.The BP-JTM4 prediction model proposed in this paper considering multi-regional and multi-meteorological factors has excellent performance. When the model parameters are optimal, RMSE and MAE can even reach 6.2457 ns and 5.0817 ns, and the MAXE is less than 15 ns, which fully reflects its superior propagation delay fitting ability. The modeling method can accurately predict the ∆*T_p_* of ASF grid vertices in the future so as to meet the *T_p_* correction requirement of the eLoran system.

## 6. Conclusions

Meteorological change is the key factor affecting the propagation delay fluctuation of the eLoran groundwave signal. In this paper, the direct and indirect disturbance effects of meteorological factors on propagation delay and the compensation methods are studied. First, the theoretical influences of relevant meteorological factors on the calculation of propagation delay components such as PF, SF and ASF are discussed, respectively. Then, through the joint correlation analysis of the multi-point measured data, the complex nonlinear relationship between the propagation delay and seven major meteorological factors is demonstrated. On this basis, the propagation delay prediction model of multi-dimensional input factors considering the regional changes of multi-meteorological factors is established based on the BPNN principle. Finally, the performance comparisons of various models show that the proposed model can effectively predict the fluctuations of propagation delay days in the future, and its overall performance is significantly improved compared with other models established in this paper and previous research results.

In addition, the study in this paper still has some limitations and extensibility, which require long-term exploration and continuous human and financial support in the future, which will also contribute to the development of meteorological forecasting and geological surveying.
Support of more detailed modeling conditions, such as a higher sampling rate of meteorological data, a longer and more complex path environment and more meteorological factors.Inversion research of geodetic conductivity based on the eLoran propagation time delay fluctuation and the development of the dynamic geodetic conductivity electronic map.

## Figures and Tables

**Figure 1 sensors-23-05176-f001:**
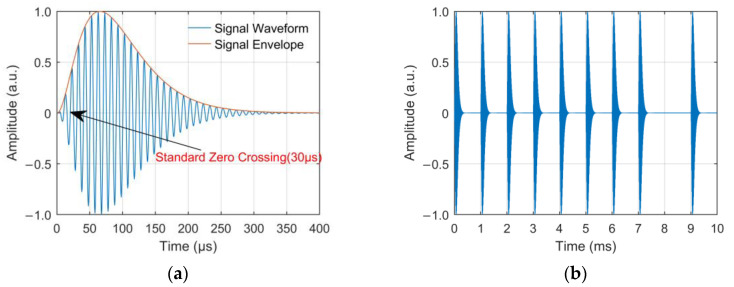
Signal waveforms of (**a**) standard Loran-C pulse; (**b**) Loran-C main station pulse group.

**Figure 2 sensors-23-05176-f002:**
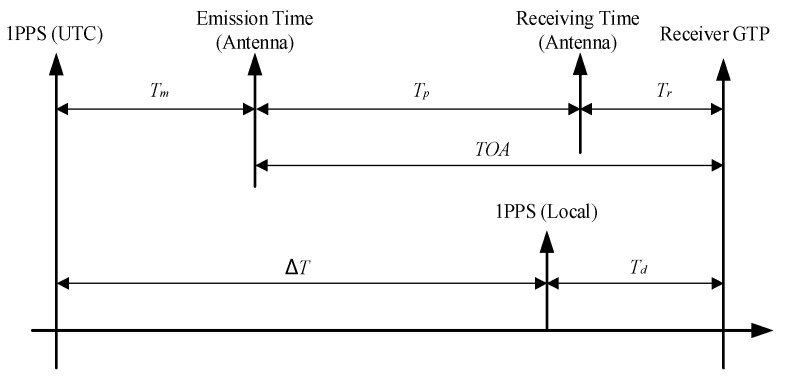
Principle of autonomous timing of the eLoran system.

**Figure 3 sensors-23-05176-f003:**
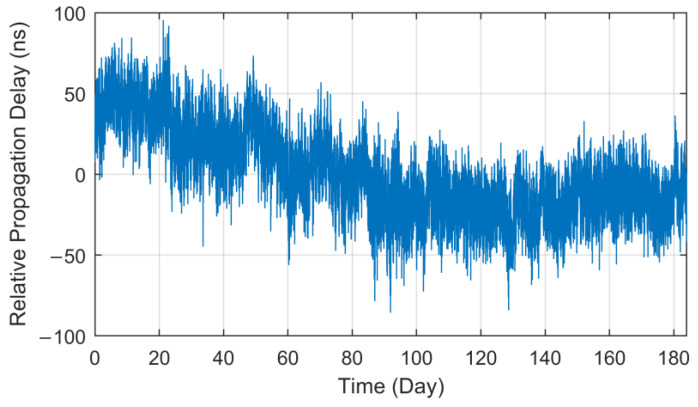
Long-term measurement value of propagation delay at the Lintong monitoring station.

**Figure 4 sensors-23-05176-f004:**
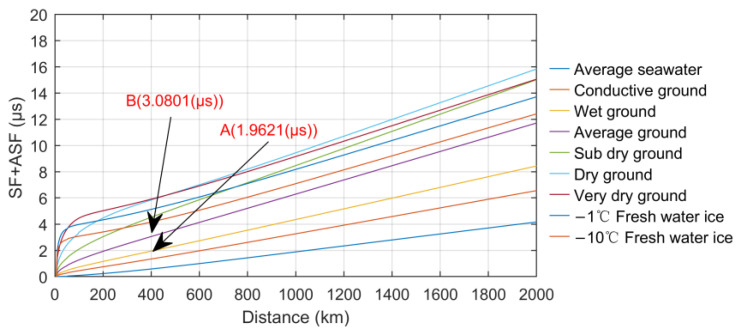
Changes in SF + ASF under different ground types.

**Figure 5 sensors-23-05176-f005:**
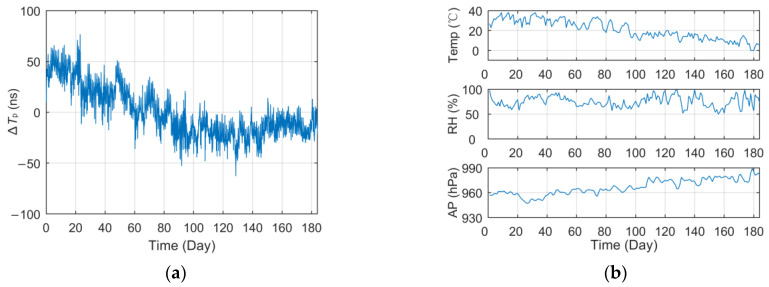
Long-term data curve and meteorological factors change of the corresponding test site (Lintong): (**a**) Relative propagation delay; (**b**) Major meteorological factors.

**Figure 6 sensors-23-05176-f006:**
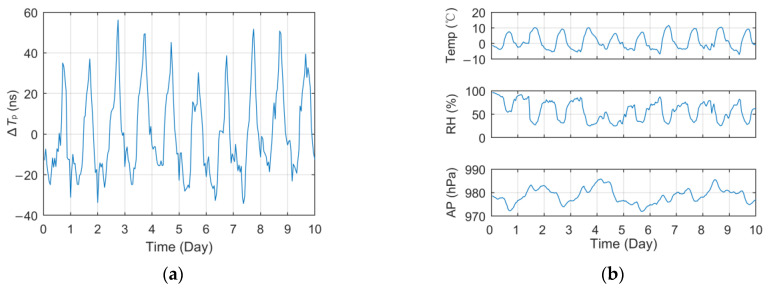
Short-term data curve and meteorological factors change of the corresponding test site (Jingyang): (**a**) Relative propagation delay; (**b**) Major meteorological factors.

**Figure 7 sensors-23-05176-f007:**
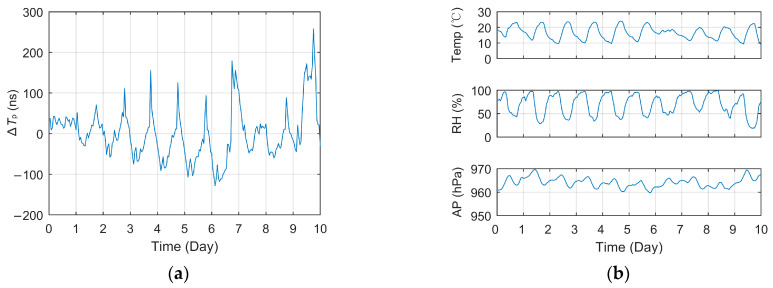
Short-term data curve and meteorological factors change of the corresponding test site (Meixian): (**a**) Relative propagation delay; (**b**) Major meteorological factors.

**Figure 8 sensors-23-05176-f008:**
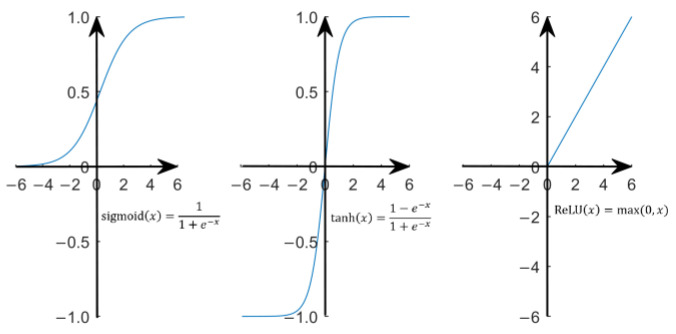
Three common activation functions.

**Figure 9 sensors-23-05176-f009:**
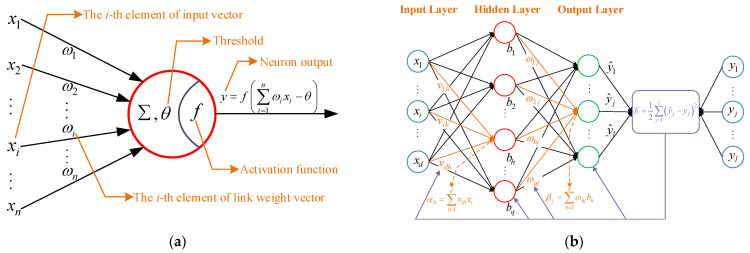
Schemes diagrams of: (**a**) Artificial neuron model; (**b**) Single-hidden layer BPNN model.

**Figure 10 sensors-23-05176-f010:**
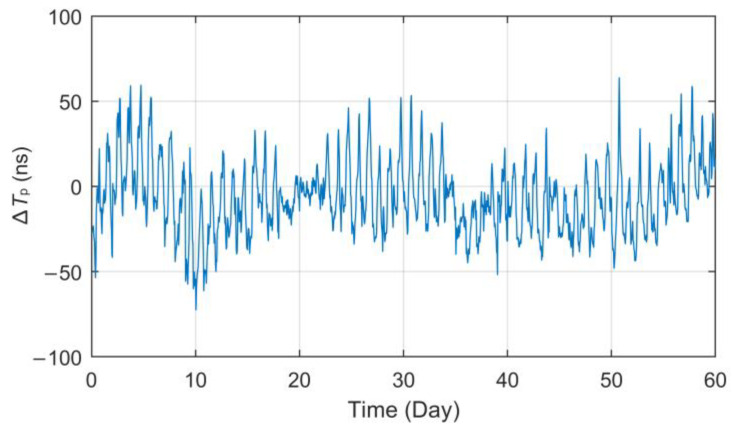
Variation curve of ∆*T_p_* at the Jingyang test site for 60 consecutive days.

**Figure 11 sensors-23-05176-f011:**
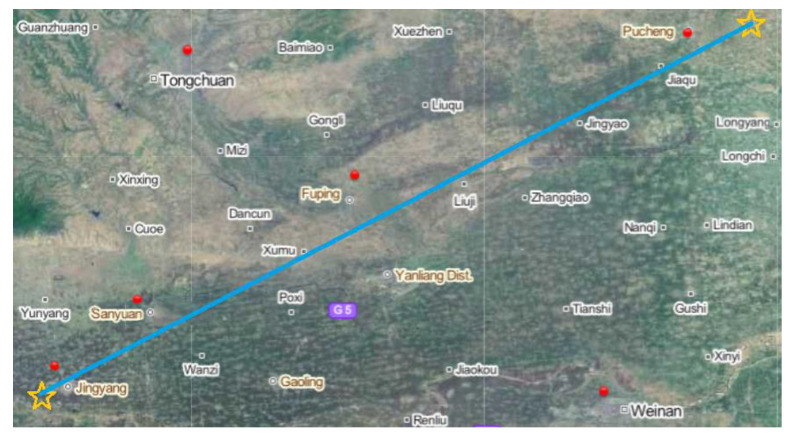
Signal propagation path and distribution of meteorological stations.

**Figure 12 sensors-23-05176-f012:**
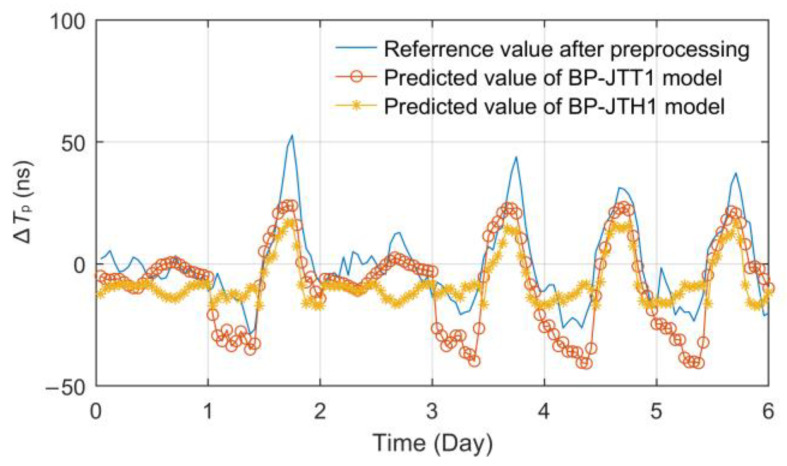
Comparison of single-factor BPNN prediction models.

**Figure 13 sensors-23-05176-f013:**
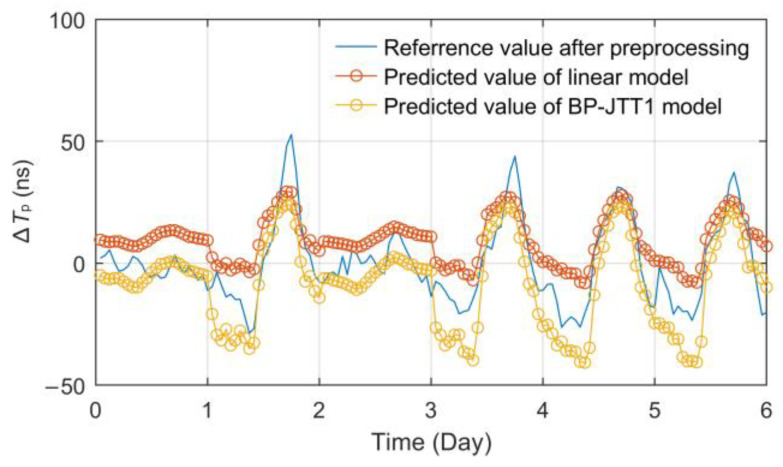
Effect diagram of the temperature linear prediction model.

**Figure 14 sensors-23-05176-f014:**
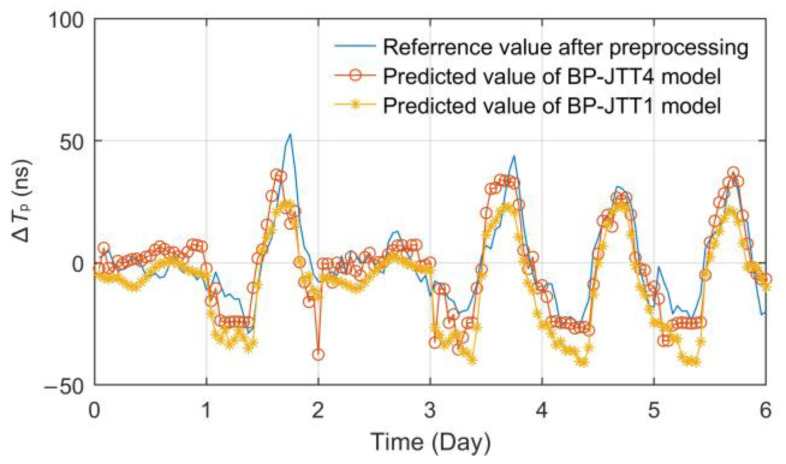
Effect diagram of the BP-JTT4 prediction model.

**Figure 15 sensors-23-05176-f015:**
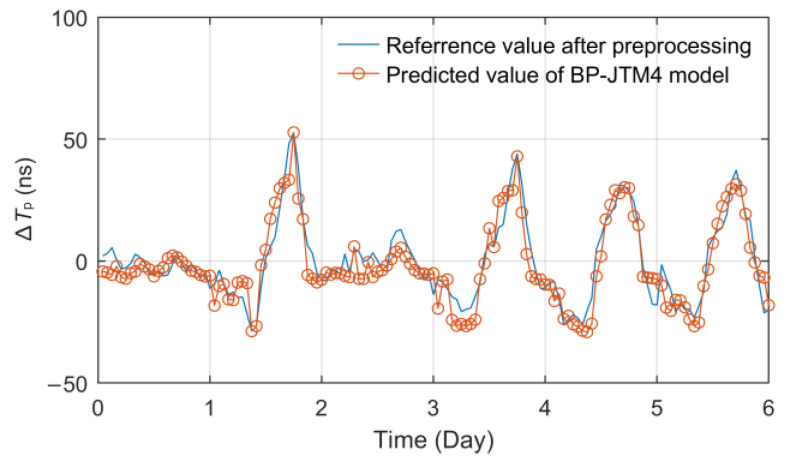
Effect diagram of the BP-JTM4 prediction model.

**Table 1 sensors-23-05176-t001:** Seasonal variation of refractive index value (unit: N) in different regions of China.

Region	January	April	July	October	Annual Average	Annual Range
Guangzhou	323	362	380	353	353	57
Shanghai	316	333	383	342	343	67
Wuhan	314	334	383	337	342	69
Beijing	301	301	361	311	320	60
Chengdu	303	320	351	320	320	48
Urumqi	292	281	290	285	285	11
Kunming	259	270	299	280	279	40
Lanzhou	253	246	272	265	278	26
Hohhot	277	260	292	272	272	32
Lhasa	192	194	220	202	202	28

**Table 2 sensors-23-05176-t002:** Conductivity parameter values of nine typical ground types.

Reference Ground Type	ε	Nominal Value of σ (S/m)	Variation Range of σ (S/m)	Equivalent Earth Radius Coefficient
Average seawater	70	5	7~3	1.14
Conductive ground	40	3 × 10^−2^	5.5 × 10^−2^~1.7 × 10^−2^	1.13
Wet ground	30	1 × 10^−2^	1.7 × 10^−2^~5.5 × 10^−3^	1.11
Average ground	22	3 × 10^−3^	5.5 × 10^−3^~1.7 × 10^−3^	1.08
Sub dry ground	15	1 × 10^−3^	1.7 × 10^−3^~5.5 × 10^−4^	1.06
Dry ground	7	3 × 10^−4^	5.5 × 10^−4^~1.7 × 10^−4^	1.05
Very dry ground	3	1 × 10^−4^	1.7 × 10^−4^~5.5 × 10^−5^	1.06
−1 °C Fresh water ice	3	3 × 10^−5^	5.5 × 10^−5^~1.7 × 10^−5^	1.06
−10 °C Fresh water ice	3	1 × 10^−5^	1.7 × 10^−5^~5.5 × 10^−6^	1.07

**Table 3 sensors-23-05176-t003:** Correlation coefficients between long-term *T_p_* and meteorological factors.

CT	TEMP	RH	AP
Pearson	0.8119	−0.0051	−0.7087
Spearman	0.7939	−0.0119	−0.7218
Kendall	0.5892	−0.0023	−0.4881

CT = correlation type; TEMP = temperature; RH = relative humidity; AP = atmospheric pressure.

**Table 4 sensors-23-05176-t004:** PCC between measured diurnal *T_p_* and meteorological factors (Jingyang and Meixian).

Test Time	Jingyang			Meixian		
	TEMP	RH	AP	TEMP	RH	AP
Day 1	0.9206	−0.7002	0.3093	0.0042	0.0188	−0.1449
Day 2	0.8996	−0.8645	−0.8277	0.7394	−0.6554	−0.7025
Day 3	0.9354	−0.9065	0.3300	0.7136	−0.6240	−0.7472
Day 4	0.9501	−0.9330	−0.4682	0.7292	−0.6936	−0.7937
Day 5	0.9539	−0.9512	−0.6129	0.8416	−0.7745	−0.7364
Day 6	0.8771	−0.8401	0.0435	0.7487	−0.6551	−0.7997
Day 7	0.9072	−0.9134	−0.8320	−0.5730	0.4745	0.1897
Day 8	0.9082	−0.9187	−0.0688	0.0790	0.1443	−0.5147
Day 9	0.9523	−0.9409	−0.5955	0.7092	−0.7936	−0.3584
Day 10	0.8864	−0.9143	−0.5088	0.8374	−0.8578	0.3577
Mean Value	0.9191	−0.8883	−0.3231	0.4829	−0.4416	−0.4250

TEMP = temperature; RH = relative humidity; AP = atmospheric pressure.

**Table 5 sensors-23-05176-t005:** Correlation coefficient between *T_p_* and various meteorological factors of each meteorological station along the propagation path.

MS (Number)	CT	TEMP	RH	AP	VP	AoP	WS	WD
Pucheng (53948)	$\rho_{XY}$	0.6225	−0.6005	−0.1822	−0.2109	0.0929	0.2025	−0.1002
	$\rho_{s}$	0.5872	−0.5912	−0.1573	−0.1762	0.1578	0.2118	−0.1061
	τ	0.4691	−0.4621	−0.1298	−0.1446	0.1205	0.1444	−0.0739
Fuping (57042)	$\rho_{XY}$	0.6145	−0.5970	−0.1802	−0.2342	0.1177	0.1232	−0.2123
	$\rho_{s}$	0.5845	−0.5738	−0.1597	−0.1783	0.1161	0.1134	−0.1801
	τ	0.4638	−0.4428	−0.1311	−0.1447	0.0858	0.0734	−0.1194
Sanyuan (57041)	$\rho_{XY}$	0.6256	−0.6222	−0.1880	−0.2219	0.1428	0.1372	−0.2939
	$\rho_{s}$	0.5931	−0.5918	−0.1627	−0.1840	0.1143	0.0980	−0.2671
	τ	0.4724	−0.4621	−0.1325	−0.1483	0.0819	0.0647	−0.1804
Jingyang (57033)	$\rho_{XY}$	0.6201	−0.5681	−0.1863	−0.2125	0.0696	0.2747	−0.1249
	$\rho_{s}$	0.6001	−0.5505	−0.1582	−0.1760	0.0516	0.2577	−0.1261
	τ	0.4786	−0.4318	−0.1293	−0.1463	0.0395	0.1797	−0.0877

MS = meteorological station; CT = correlation type; TEMP = temperature; RH = relative humidity; AP = atmospheric pressure; VP = vapor pressure; AoP = amount of precipitation.

**Table 6 sensors-23-05176-t006:** Performance indexes of propagation delay prediction models.

Meteorological Data Type	Data Scope	Model	RMSE (ns)	MAE (ns)	MAXE (ns)
RH	Single region	BP-JTH1	12.7932	13.1453	30.1544
TEMP	Single region	BP-JTT1	9.4467	10.2120	28.8836
TEMP	Single region	Linear	9.2591	11.2292	35.9205
TEMP	Full path	BP-JTT4	8.5696	6.4151	23.8801
TEMP, RH, AP, VP, AoP, WS and WD	Full path	BP-JTM4	6.2457	5.0817	14.8372

TEMP = temperature; RH = relative humidity; AP = atmospheric pressure; VP = vapor pressure; AoP = amount of precipitation.

**Table 7 sensors-23-05176-t007:** Parameters of different BP neural network models.

Model	Number of			Activation Function	Learning Rate	Iteration Times
	Input Layer Nodes	Hidden Layer	Hidden Layer Nodes			
BP-JTH1	1	1	4	Sigmoid	0.1	5000
BP-JTT1	1	1	4	Sigmoid		
BP-JTT4	4	2	10, 4	Sigmoid, Tanh		
BP-JTM4	28	2	10, 4	Sigmoid, Tanh		

## Data Availability

The data underlying the results presented in this paper are not completely publicly available at this time but may be obtained from the authors upon reasonable request.

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
