# Peer review of "ELoran Propagation Delay Prediction Model Based on a BP Neural Network for a Complex Meteorological Environment"

_sensors, 2023, doi:10.3390/s23115176_

Round 1
Reviewer 1 Report
This work proposed a propagation delay prediction model based on Back-Propagation neural network (BPNN) for complex meteorological environment. The topic of this work is interesting for the readers. The work is rather simple, but provides clear and valuable results. I believe this manuscript is suitable for acceptance in the journal with minor revisions.
1. The definitions of the activation functions (e.g., Sigmoid and Tanh) need to be provided.
2. How did you determine the learning rate of the BPNN model?
3. The results showed the BP-JTM4 model which considers multiple input features has the highest accuracy. It should be noted that there are many machine learning models. However, the authors failed to compare the BP-JTM4 model with some commonly used machine learning models such as random forest and support vector regression to further demonstrate the effectiveness of the proposed BPNN model. Please refer to the following references to compare the prediction accuracy of the BPNN model and other conventional machine learning models with the same input features. At least, the authors should discuss the advantages of the proposed BPNN model over other methods.
https://doi.org/10.1111/ffe.14032
https://doi.org/10.1016/j.ijfatigue.2020.105941
Minor editing of English is required.
Reviewer 2 Report
The main contributions of the present study are (a) the effect of meteorological factors on different longwave propagation delay components is theoretically analyzed quantitatively or qualitatively. (b) the determination of important meteorological modeling parameters. (c) establishing a prediction model based on Back-Propagation Neural Network (BPNN), to improve the accuracy of propagation delay prediction.
This is an interesting effort which leads to useful results that are publishable. Improvements based on the revisions mentioned just below are necessary:
Atmospheric sounding: Rawinsonde is an upper-air sounding that includes determination of wind speeds and directions, by tracking a balloon-borne radiosonde with a radio direction detector (GPS or Loran radio navigation signals). In the Introduction the contribution of Sir John Houghton should be mentioned (a summary is given in the editorial: DOI: 10.1080/2150704X.2021.1881649)
Temporal evolution of meteorological parameters: short term, long-term and abrupt changes in meteorological factors (such as temperature, humidity and precipitation) should be discussed based on earlier publications in the field. Special attention should be given in nonlinearities and long-range correlations (see: DOI10.1007/s00704-012-0605-2, AND https://doi.org/10.1080/2150704X.2022.2091963
Contribution to the SDGs: Please add a very brief argument on the contribution of your study to the success of the Sustainable Development Goals-UN 2030 agenda. (a summary is given in the editorial: https://doi.org/10.1080/2150704X.2020.1753338)
By summarizing the main idea of this paper is interesting and the revisions will enhance readers’ attraction
I am looking forward to read the revised version.
1. The main question addressed by the research is the establishment of a prediction model based on Back-Propagation Neural Network, to improve the accuracy of propagation delay prediction in a complex meteorological environment.2. The topic is considered original addressing the gap of the delay of a signal propagated in a complicated meteorological environment
3. The outcome of this research is the improvement of the accuracy of propagation delay prediction.
4. The methodology used is sufficient and no further controls are required.
5. The conclusions are reasonably consistent with the evidence and arguments presented and adequately address the main question posed.
6. The references are quite appropriate. However, additional references are required from previously published studies, notably:
---Atmospheric sounding: Rawinsonde is an upper-air sounding that includes determination of wind speeds and directions, by tracking a balloon-borne radiosonde with a radio direction detector (GPS or Loran radio navigation signals). In the Introduction the contribution of Sir John Houghton should be mentioned (a summary is given in the editorial: DOI: 10.1080/2150704X.2021.1881649)
---Temporal evolution of meteorological parameters: short term, long-term and abrupt changes in meteorological factors (such as temperature, humidity and precipitation) should be discussed based on earlier publications in the field. Special attention should be given in nonlinearities and long-range correlations (see: DOI10.1007/s00704-012-0605-2, AND https://doi.org/10.1080/2150704X.2022.2091963
---Contribution to the SDGs: Please add a very brief argument on the contribution of your study to the success of the Sustainable Development Goals-UN 2030 agenda. (a summary is given in the editorial: https://doi.org/10.1080/2150704X.2020.1753338)
7. No comments on Tables and Figures
see above
